# Effects of the Musical Sound Environment on Communicating Emotion

**DOI:** 10.3390/ijerph17072499

**Published:** 2020-04-06

**Authors:** Qi Meng, Jiani Jiang, Fangfang Liu, Xiaoduo Xu

**Affiliations:** 1Key Laboratory of Cold Region Urban and Rural Human Settlement Environment Science and Technology, Ministry of Industry and Information Technology, School of Architecture, Harbin Institute of Technology, 66 West Dazhi Street, Nan Gang District, Harbin 150001, China; mengq@hit.edu.cn (Q.M.); 18S134156@stu.hit.edu.cn (J.J.); 2UCL The Bartlett School of Architecture, University College London (UCL), London WC1H 0QB, UK

**Keywords:** music, sound environment, communicating emotion, PAD, social characteristics

## Abstract

The acoustic environment is one of the factors influencing emotion, however, existing research has mainly focused on the effects of noise on emotion, and on music therapy, while the acoustic and psychological effects of music on interactive behaviour have been neglected. Therefore, this study aimed to investigate the effects of music on communicating emotion including evaluation of music, and d-values of pleasure, arousal, and dominance (PAD), in terms of sound pressure level (SPL), musical emotion, and tempo. Based on acoustic environment measurement and a questionnaire survey with 52 participants in a normal classroom in Harbin city, China, the following results were found. First, SPL was significantly correlated with musical evaluation of communication: average scores of musical evaluation decreased sharply from 1.31 to −2.13 when SPL rose from 50 dBA to 60 dBA, while they floated from 0.88 to 1.31 between 40 dBA and 50 dBA. Arousal increased with increases in musical SPL in the negative evaluation group. Second, musical emotions had significant effects on musical evaluation of communication, among which the effect of joyful-sounding music was the highest; and in general, joyful- and stirring-sounding music could enhance pleasure and arousal efficiently. Third, musical tempo had significant effect on musical evaluation and communicating emotion, faster music could enhance arousal and pleasure efficiently. Finally, in terms of social characteristics, familiarity, gender combination, and number of participants affected communicating emotion. For instance, in the positive evaluation group, dominance was much higher in the single-gender groups. This study shows that some music factors, such as SPL, musical emotion, and tempo, can be used to enhance communicating emotion.

## 1. Introduction

Emotions are, in effect, organized cognitive-motivational-relational configurations whose status changes with changes in the person-environment relationship as this is perceived and evaluated. Emotion is closely related to psychological and somatic health [1,2] and may lead to heart diseases [3,4], hypertension, diabetes [5], and even non-suicidal self-injury [6]. On the other hand, in social interaction, social behaviour is constantly guided by cues that we interpret about the emotions of others [1,7]. Furthermore, interpersonal relationships can be fulfilled by emotions [8]. The concept of emotional contagion means that there is ripple effect (which is also known as an effect of imitation, proposed by Kounit) among human interaction through conscious or unconscious induction of emotional states and behavioural attitudes [9]. Negative emotions, especially anger, can easily lead to violence [10,11], whereas in positive emotions, positive correlations have been found between total amount of face-to-face interaction and mood [12]. Communication is a fundamental part of social face-to-face interaction [13] that can produce cooperation or coordination. For instance, communicating disappointment in the other is conducive to establishing cooperation when the interaction partner does not return a favour [8]. However, how to balance communication emotions through regulating the sound environment has been seldom studied.

Previous studies have shown that emotion can be affected by environmental contexts [1,14], including luminous, thermal, and acoustic factors. In terms of the luminous environment, some studies have found that increasing indoor illumination sufficiently can help treat seasonal affective disorder [15]. On the other hand, a cold or hot environment can elicit negative emotions, which are distracting and reduce performance efficiency [16]. As for the acoustic perception, the term ‘soundscape’ was coined by Schafer, who defined it as a sonic environment, with an emphasis on the way it is perceived and understood by individuals or society [17]. In 2014, the International Organization of Standardization developed a broader definition of soundscape: acoustic environment as perceived or experienced and/or understood by a person or people, in context [18]. Assessment of soundscape is part of sensory aesthetics research and the aesthetic response of surroundings is considered to be a mix of high pleasure, excitement and relaxation [19,20]. In terms of positive effects, a study suggested that nature- and culture-related sounds, such as fountains, birds singing, bells and music from clock, which induce tranquil and pleasant feelings, are preferred in urban squares, as opposed to artificial sounds [21]. On the other hand, there are also many studies about the negative emotions induced by noise. For instance, some studies suggest that the relationship between annoyance and noise is also related to some psychoacoustic indices, such as loudness, sharpness, roughness, impulsiveness, and so on [22]. Additionally, a growing body of research indicates that it is not the physical properties of sound, but the message conveyed within the sound that leads to different emotions. For instance, a related study found that auditory cortex would encode information on the emotional-motivational valence of sounds so that people could recognize sound as aversive or pleasant [23]. Therefore, using emotional sounds in a soundscape design is an effective way to regulate emotion.

On the other hand, music, as a special kind of acoustic stimulus, is one of the factors affecting emotion, which has been widely applied in film, marketing, and therapy [24]. For example, for patients, music can decrease depression [25], anxiety, and stress [26]. Evidence against a strict cognitivist position suggests that music can induce some sort of an emotional response [27,28]. There are nine common emotions (wonder, transcendence, tenderness, sadness, nostalgia, peacefulness, power, joyful activation, and tension) evoked by music [29].

Existing findings indicate that tempo, sound pressure level (SPL), and musical emotions may also influence emotion [27], which have been assessed using behavioural, physiological, and neurological measures [30]. In terms of tempo, studies have found that fast tempo background music can lead to higher arousal than slow tempo background music, which would affect spending behaviour [31,32], and fast tempo music can increase the arousal level of brain activity [33,34]. In terms of SPL, music with high SPL can increase perceived activation and tension [35]. In terms of musical emotion, a significant body of research has found that the relationships between the emotion perceived and the emotion felt in music do not always match. In general, music can arouse emotions similar to the emotional quality perceived in music, whereas sometimes, fearful music is perceived as negative but felt as positive [36,37]. Besides, the emotion felt are frequently rated same as or lower than the emotion perceived, which may be attributed to inhibition of emotion contagion [38]. Studies have mainly focused on how listeners evaluate musical emotions when listening to music on purpose [28,39]. However, an individual listens to music in isolation and, even when listening occurs in a social setting, it may not co-occur with social interactions; the combined effects of music and social interaction on communicating emotions have been less studied.

How we respond to music is not only affected by musical properties, but also listener properties [36]. Studies have found that social characteristics, such as gender, age, and musical education [27], can lead to different emotion evaluations. In terms of gender, the preferred SPL with earphones for males was 5.4 dBA higher than that for females [40]. In terms of age, compared to younger children, older listeners can recognize the perceived emotion in music more correctly. Besides, older listeners hold more emotional experiences, and music sometimes would revoke their personal and private memories which may also lead to different felt emotions [38,41]. In terms of musical education, studies show that musicians have better emotional recognition ability than do non-musicians [42,43]. Additionally, influential factors include situation, such as being alone or having a company [36]; thus, it is necessary to consider the number of participants involved in communication.

Therefore, this study aimed to investigate the effects of music on communicating emotion by rating how participants felt before and after the conversation based on the following research questions. First, is SPL of music one of the main factors contributing to communicating emotion, and if so, what is the extent of its effect and is there a best SPL during the communicating process? Second, how does musical emotion affect communicating emotion? What is the most appropriate emotional music for communication? Third, does music tempo affect communicating emotion, and if so, at what level does it have an optimal effect? Fourth, do social characteristics, including gender combination, familiarity, and number of participants, affect emotional evaluations during communication? Using an acoustic environment measurement and a questionnaire survey with 52 participants in a normal classroom in Harbin City, China, this study aimed to explore the effects of music in terms of SPL, musical emotion and tempo and evaluation of music on communicating emotion–specifically, d-values of pleasure, arousal, and dominance (PAD).

## 2. Material and Methods

### 2.1. Experimental Site

Related emotional experiments are usually conducted in the field or a laboratory, of which field experiments are carried out in naturally-occurring settings that may have high reliability and realism [44,45]. However, compared to laboratory experiments, there are some shortcomings to field experiments, such as the relative difficulty of replication and randomization bias [46]. Furthermore, there is no clear definition of communication space; therefore, there are no specific requirements of experienced subjects and case sites. Considering the controllability of music and the unpredictability of the surrounding people, a laboratory experiment was chosen for this study.

In terms of the scale of the experimental room, network surveys show that the size of a typical communication space, such as a cafe, is from 10 m^2^ to 600 m^2^. An ordinary classroom (7.5 m × 6 m × 3.5 m) in the School of Architecture at the Harbin Institute of Technology within the above range was selected as the experimental site. The indoor layout of the experimental room is shown in Figure 1. The experimental room was made of white brick wall, terrazzo floor, and a concrete ceiling. There were two glass windows, one door, 10 plywood tables, and 22 upholstered chairs in the room. The sound source was located on the table next to the window. Participants communicated in the space of a red dashed box, and an acoustic measuring point was located at the red dot shown in Figure 1a. During the whole process of the experiment, the door and windows remained closed. Considering the potential effect of tension, snacks and drinks were offered to create a relaxed communication atmosphere.

In the present study, indoor temperature and illuminance were considered so that they would not influence emotion and performance [47,48]. In terms of thermal factors, related experimental results showed that performance improves as the indoor temperature reaches 23 °C [49], and then decreases when the temperature rises above 25 °C [50]. Therefore, the experiment time was set from 9 a.m. to 4 p.m. to maintain the indoor temperature at 23 to 25 °C.

In terms of luminous factors, previous studies showed that illuminance and light colour could affect fatigue and emotion [51,52]. Preferred illuminance was examined from 310 to 600 lux, among which 500 lux was the most appropriate [53,54,55]. As for light colour, a neutral white (4000 K) space was perceived as best [55]. To decrease the effects of light environment, neutral white (4000 K) light was used in this study, and during the experiment lamps in the room maintained an average illuminance of 480.5 lux.

Reverberation time (RT) is another factor influencing speech intelligibility [56,57], which may influence the quality of communication. RT can be calculated by specific data concerning indoor materials, including absorption coefficient, area, and amount. Based on the Elaine formula, the absorption values of specific materials in the experiment room were computed. When the number of participants in communication changed from 2 to 4, the RT of the room changed from 0.59 s to 0.6 s, which was close to the best RT (0.6 s) according to the standard of Guidelines for Community Noise, meeting the requirement of speech intelligibility.

### 2.2. Participants

To ensure adequate statistical power, G*Power (a general power analysis program) was used to analyse the minimum sample size of subjects, assuming an effect size of d = 0.5, α = 0.05, Power (1-β) = 0.8. For the main research questions regarding to the effects of SPL, musical emotions, and tempo on communicating emotions, the minimum required sample size was 40 in the one-way ANOVA (Analysis of Variance) (within-group). Therefore, in this study, 52 participants were randomly sampled from among 500 college students. Of them, there were 26 females (average age = 23.85, SD = 1.73), and 26 males (average age = 23.96, SD = 1.48). All participants were in good health and had normal hearing upon inspection. Social characteristics of the participants in terms of gender, familiarity and number of participants are shown in Table 1. Single gender means that all participants were either male or female, while mixed gender means that the sample consisted of both males and females.

To avoid potential confounding effects of formal training in music [36], all participants were non-music majors and had never undergone formal training. Before the experiment, participants were required to maintain their emotional stability, obtain sufficient sleep, and follow the same routine used during the survey period, without any interference from other events, such as exams or parties. This study was reviewed and approved by the university’s institutional review board. Written informed consent was obtained from all participants. As incentives for participation, participants received either a monetary reward or various learning materials.

### 2.3. Questionnaire Survey

A subjective questionnaire was used in this study to evaluate participants’ emotional state with a high level of reliability, based on related studies indicating that self-reports seemed to be the best and most natural method to study emotional responses to music [36].

In terms of emotional evaluation, dimensional structures have been proposed to portray how emotional responses are organized psychologically [1,58,59]. Among these, the three-dimensional theory of emotion—PAD (pleasure-arousal-dominance) [60,61,62]—has been widely applied, and was chosen in this study. The questionnaire consisted of three parts. Specific subjective evaluation descriptions are shown in Table 2.

First, information about basic social characteristics and situations was collected, including gender combination, age, familiarity with communication objects, and the number of participants in communication. Second, musical evaluation during the process of communication was rated on a 21-point scale (0: no effect/1 to 10: positive/−1 to −10: negative). Third, emotional evaluation in terms of pleasure, arousal, and dominance, before and after communication, was each rated on a 10-point scale.

### 2.4. Experimental Materials

To determine how music affects emotion in terms of SPL, musical emotion, and tempo, it was necessary to address three questions: first, what kinds of musical emotions to choose and the specific excerpts to select; second, how to set different tempos of music; third, the range and intervals of musical SPL.

#### 2.4.1. Musical Emotion

There have been significant attempts at characterizing musical emotions perceived by listeners and describing the affective value and the expressiveness of music [39,63]. Both discrete classification model and continuous dimensional model are commonly used for musical emotion classification [63]. In terms of discrete classification, based on the compilation of music-relevant affect terms, adjectives are classified into groups, each group is treated as a whole, representing one type of feeling tone. For instance, Hevner presented eight synonym clusters to describe emotional perception of music, including happiness, gracefulness, sereneness, dreaminess, sadness, dignity, vigorousness and excitement [39]. Meanwhile, musical emotions can be described as vectors; typical dimension models include valence-arousal model (VA) and pleasant-arousal-dominance (PAD) models. Further, musical emotion characteristics can be classified based on musical dimensions, such as pitch height, loudness, timbre, tempo, and intensity [64,65]. For instance, referring to the Plutchik 3-D Circumplex Model, a pleasure-arousal (PA) model of musical emotion classification based on register, tempo and dynamic was presented [66].

The level of emotional perception of music varies in different musical emotions. In some studies of musical correlates for different emotions, judgments of happiness and sadness tend to be more consistent than for other emotions, such as fear and anger [27]. Moreover, some studies have shown that musical excerpts could be easily discriminated along the relaxing-stimulating and the pleasant-unpleasant dimensions [58]. Considering the diversity and consistent judgement of musical emotions, the following emotions were chosen in this study: joyful-, stirring-, tranquil- and sad-sounding music. In terms of music content, subjects’ familiarity with the music and whether music with lyrics would affect emotion [27,67] are relevant factors. There are several established theoretical models, among which a neural network audio classification scheme has been widely applied [68,69].

In this study, 12 music excerpts of high familiarity without lyrics were chosen from a collection based on RGEP (gene expression programming) [70]. In the pre-experiment, 32 college students evaluated the musical emotion of these 12 music excerpts. There were 30 valid responses; the related results are shown in Table 3.

The data showed the highest agreement of musical emotion evaluation among these 12 music excerpts of four musical emotions respectively: Joyful–83.33%, Spring Festival Prelude; Stirring–80%, Athletes March; Tranquil–80%, A Comme Amour; and Sad–90%, Butterfly Lovers.

Numerous experiments on the psychology of music have found considerable high consistency in emotional judgments of music [28]. The specific musical emotions are perceived more correctly when the degree of agreement is higher. Therefore, these four music excerpts were chosen as experimental materials to investigate how these specific perceived emotions in music excerpts would affect participants’ communicating emotion unconsciously.

#### 2.4.2. Tempo

Tempo variation has consistently been associated with differential emotional response to music [71]. Some studies indicate that happy/sad reactions can be influenced by tempo, and ratings of happiness increase as the tempo of music increases [72]. Another study showed that faster tempo is associated with stronger happiness and anger [73]. Tempo can be classified based on beats per minute (BPM) into three types: slow (0 < BPM value ≤ 100), medium (100 < BPM value ≤ 135), and fast (BPM value > 135) [74]. A metronome was used to analyse the tempo of the four music excerpts selected in 2.4.1. The music excerpts had the following BPM values: Spring Festival Prelude was 144 (allegro), Athletes March was 110 (allegretto), A Comme Amour was 70 (andante), and Butterfly Lovers was 50 (adagio). Additionally, the melodic structures of the chosen music excerpts are generally regular, with little tempo fluctuations, therefore, the average BPM could be used in tempo analysis. In the following sections, the tempo is divided into three groups for comparison, namely slow tempo (A Comme Amour, and Butterfly Lovers), medium tempo (Athletes March) and fast tempo (Spring Festival Prelude).

#### 2.4.3. SPL of Music

In terms of SPL of music, based on the data of average background SPL with participants talking from a previous study that was around 50 dBA [75], and the significant difference of 5 dBA [76], the SPL settings of music in this study were 40 dBA, 45 dBA, 50 dBA, 55 dBA, and 60 dBA, with intervals of 5 dBA. Additionally, in terms of musical emotion, related studies showed that peacefulness is a complex emotion that overlaps with each quadrant of a circumplex model defined by the dimensions of arousal and valence [27] without clear emotional directivity compared to happiness, excitement, and sadness. Therefore, after comprehensive consideration, tranquil-sounding music was chosen to examine the effects of musical SPL, and there were eight music excerpts to prepare: A Comme Amour (40 dBA to 60 dBA), Spring Festival Prelude (50 dBA), Athletes March (50 dBA), and Butterfly Lovers (50 dBA).

The sound level meter (BSWA801, BSWA, Beijing, China) was located at the measuring point shown in Figure 1 and was used to provide multiple measures of background SPL for 1 min without subjects, and under the conditions without and with music. A regular polyhedron directionless loudspeaker was used to play music. The software Adobe Audition (Adobe, San Jose, USA) was used to alter the SPL of music to meet the design value. The software SvanPC++ (SVENTEK, Poland) was used to analyse the recorded audio. After repeated measurements, the following data were obtained after summarizing: without music, the average background SPL was 32.5 ± 0.5 dBA (SD = 1.4); data of the average background SPL with music, A Comme Amour (40 dBA) was 40.2 ± 0.5 dBA(SD = 4.98), A Comme Amour (45 dBA) was 45.19 ± 0.5 dBA (SD = 5.6), A Comme Amour (50 dBA) was 49.66 ± 0.5 dBA (SD = 6.19), A Comme Amour (55 dBA) was 55.34±0.5 dBA (SD = 6.33), A Comme Amour (60 dBA) was 60.28 ± 0.5 dBA (SD = 5.9), Spring Festival Prelude (50 dBA) was 50.38 ± 0.5 dBA (SD = 5.78), Athletes March (50 dBA) was 50.31 ± 0.5 dBA (SD = 3.52), and Butterfly Lovers (50 dBA) was 50.06±0.5 dBA (SD = 5.25). Since all of the measured musical SPL data were close to the experimental set value of SPL, the following data are denoted as 40 dBA, 45 dBA, 50 dBA, 55 dBA, and 60 dBA in this study.

### 2.5. Experimental Design

This study hypothesized that musical properties, such as SPL, musical emotions and tempo, would affect communicating emotions unconsciously when participants perceived the music. Thus, the experiment was divided into eight groups, using a control variable method. Each experiment involved exposure to each level of variables in a within-group design. For example, when the specific variable was SPL, musical emotion and tempo remained unchanged. In terms of experimental means, random assignment, which has high internal validity [77], was used to eliminate systematic differences between the treatment and control groups. A total of 52 participants were randomly allocated into groups of 2–4 people to communicate every time before each experiment. In terms of the duration of communication, existing studies found that conversation duration was one of the factors affecting people’s focused attention and emotion during communication [78], and through an analysis of the average and minimum speech time of the participants, five minutes seemed appropriate for communication [79]. The flow of the experiment is shown in Figure 2.

After the random allocation, the first participants entered the experimental room and took around 2–3 min to get familiar with the experimental environment and the people they would talk to. Then, questionnaires were distributed, and participants were required to fill in their basic information according to the instructions in 2–3 min. The participants then started communication under the specific music excerpt, and kept chatting for 5 min until the music stopped. Considering variations in conversation topics, participants chose topics by themselves. Finally, participants were required to finish the rest of the questionnaires in 3 min and submitted the questionnaire, after which the experiment was completed.

### 2.6. Data Analysis

Based on collection of subjective evaluation data of questionnaires and the background SPL data through the software program SvanPC++ (SVENTEK, Poland), SPSS (Statistical Product and Service Solutions, IBM, America) was used to analyse the data from the survey. Analysis of variance (ANOVA) was used to test for significant differences under different levels of background SPL, musical emotion, tempo (within-group), and familiarity with the participants (between-group). Independent-sample t-tests were performed to assess gender differences, and the difference between one-on-one and multi-participant group. Furthermore, effect size was used to evaluate the relationship between influencing factors of musical evaluation and PAD emotion changes (pleasure, arousal, and dominance). In the ANOVA, an effect size *ƞ^2^* of 0.01 indicated a small effect; 0.06 indicated a medium effect; 0.14 indicated a large effect. In the t-test, an effect size d of 0.2 indicated a small effect; 0.5 indicated a medium effect; 0.8 indicated a large effect.

## 3. Results

As mentioned above, SPL of music, musical emotions, tempo, and social characteristics may influence communication emotions. This chapter mainly consists of four sections that analyse the relationships between these factors and percentages of musical evaluations, average evaluation scores, and d-values of PAD emotional evaluations. Section 3.1 examines the effects of SPL of music on communication emotions from 40 dBA to 60 dBA. Section 3.2 explores the influence of musical emotions on communication emotions, including joyful-, stirring-, tranquil- and sad-sounding music. Section 3.3 shows the different effects of medium and fast tempo music on communication emotion. Section 3.4 discusses the effect of social characteristics on communication emotions.

### 3.1. Effect of SPL of Music on Communication Emotion

The relationships between SPL of music from 40 dBA to 60 dBA and percentages of musical evaluations (no effect/positive/negative) as well as average scores of musical evaluations are shown in Figure 3. To determine the distribution of participants in terms of the three levels of musical evaluation—positive, negative, and no effect—percentages were also analysed. In terms of percentages of different musical evaluations, percentages of positive evaluations rose slightly from 43.1% to 48.1% from 40 dBA to 50 dBA, and when SPL reached 60 dBA, it decreased to 28.8%. As musical SPL rose, the percentage of neutral evaluations showed a great drop from 52.9% to 9.6%. One the contrary, the percentages of negative evaluations rose from 3.9% to 61.5% with SPL increasing from 40 dBA to 60 dBA. It was noticeable that musical evaluation distributions were similar under 45 dBA and 50 dBA.

In terms of average scores of musical evaluations, generally, within the SPL range of 40–60 dBA, the overall effect of music was positive (average score > 0) when SPL was less than or equal to 55 dBA; when SPL reached 60 dBA, the musical effect turned negative. Based on the five levels of SPL, the results of the ANOVA showed that SPL of music was significantly correlated with musical evaluation (df1 = 4, df2 = 254, F = 5.065, *p* < 0.01). Average scores of musical evaluations moved from 0.88 to 1.31 when SPL rose from 40 dBA to 50 dBA, while it decreased sharply from 1.31 to −2.13 when SPL exceeded 50 dBA. It is interesting to find that when SPL of music was 50 dBA, participants rated it highest (1.31) and gave it the highest percentage of positive evaluations (48.1%). The results indicated that considering both distribution and the average scores of musical evaluations, the most positive musical effect on communication was shown under 50 dBA.

Since the percentage of negative evaluations under 40 dBA was too small, relationships between musical SPL and PAD emotional evaluations were mainly analysed under 45 dBA to 60 dBA in the positive/negative groups, respectively. Based on the four levels of SPL, the results of the ANOVA showed no significant differences in the positive group, for pleasure (df1 = 3, df2 = 85, F = 0.404, *p* = 0.75), for arousal (df1 = 3, df2 = 85, F=0.666, *p* = 0.57), and for dominance (df1 = 3, df2 = 85, F = 0.337, *p* = 0.79). And there was no significance in all participants either, for pleasure (df1 = 3, df2 = 204, F = 0.183, *p* = 0.91), for arousal (df1 = 3, df2 = 201, F=2.045, *p* = 0.11), and for dominance (df1 = 3, df2 = 203, F = 0.055, *p* = 0.98). Therefore, the following focused on the data in the negative group.

The average d-values of PAD emotional evaluations under different musical SPL from 45 dBA to 60 dBA in the negative group are shown in Figure 4. The results of the ANOVA showed that the SPL had a significant effect on arousal (df1 = 3, df2 = 63, F = 3.984, *p* < 0.05), but there was no significant effect on pleasure (df1 = 3, df2 = 66, F = 0.546, *p* = 0.65, *ƞ^2^* = 0.006) and dominance (df1 = 3, df2 = 65, F = 1.149, *p* = 0.34, *ƞ^2^* = 0.001). In terms of pleasure, no matter under what SPL between 45 dBA and 60 dBA, pleasure decreased 0.88 to 2.27, among which, it was noticeable that pleasure was lowest (−2.27) under 50 dBA in the negative group, while the overall musical evaluation was highest under 50 dBA. In terms of arousal and dominance, arousal increased continuously from −1.18 to 1.94 with the increase of SPL from 45 dBA to 60 dBA. Moreover, dominance rose from −0.64 to 1.00. These findings indicate that when the musical evaluation was negative during communication, arousal could be enhanced efficiently with the increase of SPL. Similar conclusions were also drawn in previous studies that showed that musical up-ramps (60 to 90 dB SPL) elicited significantly higher ratings of emotional arousal change [80].

### 3.2. Effect of Musical Emotion on Communication Emotion

The relationships between musical emotions and percentages of musical evaluations (no effect/positive/negative) as well as average scores of musical evaluations are shown in Figure 5. In terms of percentages of different musical evaluations, percentages of positive evaluations were slightly different in these four musical emotions, which were lower (48.1%) under stirring-, and tranquil-sounding music, and higher under joyful- (55.8%) and sad-sounding (57.7%) music. It was noticeable that the percentage of positive evaluations was highest under sad-sounding music; previous studies also indicated that sad-sounding music could elicit higher levels of mixed feelings compared to joyful-sounding music [81]. Percentages of neutral evaluations were similar under joyful-, stirring-, and tranquil-sounding music, ranging from 28.9% to 32.7%, while it was lowest (23.1%) under sad-sounding music. Percentages of negative evaluations were close under stirring-, tranquil-, sad-sounding music, which ranged from 19.2% to 23.1%, while it was lowest (11.5%) under joyful-sounding music. It was noticeable that under stirring-sounding music, the percentage of positive evaluation was lowest, and at the same time, its percentage of negative evaluation was highest.

In terms of average scores of musical evaluations, generally, the overall effect of music was positive (average score > 0) under each of the four musical emotions (50 dBA). Based on the four levels of musical emotions, the results of the ANOVA showed a significant effect of musical emotion on musical evaluation for all participants (df1 = 3, df2 = 394, F = 2.885, *p* < 0.05). Among these, joyful-sounding music showed the highest evaluation (2.92), followed by sad-sounding music (1.52), tranquil-sounding music (1.31), and stirring-sounding music being the lowest (1.04). The findings indicate that among these four musical emotions, communication could be efficiently enhanced under joyful-sounding music, whereas the positive effect of stirring-sounding music was least obvious.

The average d-values of PAD emotional evaluations under different musical emotions, specifically, joyful-, stirring-, tranquil-, and sad-sounding music, in all participants and of negative group are shown in Figure 6a,b.

In all participants, generally, pleasure, arousal, and dominance all increased under these four musical emotions except for pleasure under tranquil-sounding music. The results of the ANOVA showed that in terms of pleasure, the difference between various musical emotions was significant (df1 = 3, df2 = 394, F = 2.943, *p* < 0.05). Under joyful-, stirring-, and sad-sounding music, the d-values of pleasure increased ranging from 0.17 to 0.63, while pleasure decreased 0.41 under tranquil-sounding music. Moreover, in terms of arousal, there was significant difference between various musical emotions (df1 = 3, df2 = 394, F = 3.572, *p* < 0.05). Joyful-sounding music showed the highest d-value (1.51), followed by stirring-sounding music (1.44), tranquil-sounding music (0.65), and sad-sounding music being the lowest (0.57). In terms of dominance, differences between d-values (ranging from 0.62 to 0.98) under these four musical emotions were not significant (df1 = 3, df2 = 394, F = 0.466, *p* = 0.706), and the d-value under stirring-sounding music was the lowest (0.62). The findings indicate that generally speaking, there was significant effect of musical emotions on communicating emotions, joyful- and stirring-sounding music could enhance pleasure and arousal efficiently.

In the negative group, the d-values of PAD evaluations varied widely under different musical emotions, during communication, the results of the ANOVA showed that musical emotions in terms of pleasure, the difference among four musical emotions was not significant (df1 = 3, df2 = 36, F = 1.683, *p* = 0.18, *ƞ^2^* = 0.123), among which, the differences between stirring- and tranquil-sounding music was significant (*p* < 0.05). Other than stirring-sounding music (0.54), tranquil-sounding music showed the greatest drop (−2), followed by joyful-sounding music (−1.67), and sad-sounding music (-0.4). In terms of arousal, the d-values of arousal were all above 0 under all of these four musical emotions ranging from 0.2 to 1.38 with no significant differences (df1 = 3, df2 = 35, F = 0.376, *p* = 0.77, *ƞ^2^*= 0.031). The increase in arousal was highest under stirring-sounding music (1.38) while the differences of the other three musical emotions were not significant (from 0.2 to 0.7). In terms of dominance, the difference among musical emotions was not significant (df1 = 3, df2 = 36, F = 0.543, *p* = 0.65, *ƞ^2^* = 0.043). Except for tranquil-sounding music (d-value = -0.18), dominance increased from 0.6 to 0.92 under the other musical emotions. It was noticeable that pleasure, arousal, and dominance all increased under stirring-sounding music. The results indicate that, when the musical evaluation was negative, stirring-sounding music could enhance pleasure efficiently as opposed to sad-sounding music.

While in positive group, the results of the ANOVA showed that, the difference between various musical emotions was not significant in terms of pleasure (df1 = 3, df2 = 106, F = 1.749, *p* = 0.16, *ƞ^2^* = 0.047), arousal (df1 = 3, df2 = 106, F = 1.701, *p* = 0.17, *ƞ^2^* = 0.046) and dominance (df1 = 3, df2 = 105, F = 0.645, *p* = 0.58, *ƞ^2^* = 0.018).

### 3.3. Effect of Tempo on Communication Emotion

The relationships between tempo and percentages of musical evaluations (no effect/positive/negative) as well as average scores of musical evaluations are shown in Figure 7. In terms of percentages of musical evaluations, the distribution of positive evaluations were similar, for the slow (53.9%), medium (48.1%), and fast (55.8%), 20.2%, 25.0%, and 11.5% being negative evaluation, respectively. In terms of average scores of musical evaluations, the results of the ANOVA showed that the differences between the slow, medium and fast groups were significant (df1 = 2, df2 = 395, F = 3.385, *p* < 0.05). The average scores of musical evaluation were lowest in the medium group (1.04) and highest in the fast group (2.40). The findings indicate that in terms of musical tempo, fast music played a more positive role in communication compared to music of slow- and medium-tempo music.

The average d-values of PAD emotional evaluations under different musical tempos for all participants are shown in Figure 8. Based on the three levels of tempo, the results of the ANOVA showed that for all participants, there were significant effects on pleasure (df1 = 2, df2 = 395, F = 3.329, *p* < 0.05) and arousal (df1 = 2, df2 = 395, F = 5.352, *p* < 0.01). Average d-values were the lowest in the slow tempo group for pleasure (0.11) and arousal (0.67), followed by 0.65 and 1.46 in the medium group and, 0.60, and 1.46 in the fast group, respectively. In terms of dominance, d-values of different tempos were similar; and there was no significant difference in dominance (df1 = 2, df2 = 395, F = 0.701, *p* = 0.49, *ƞ^2^* = 0.004). The findings suggest that generally, the rise of music tempo could increase pleasure and arousal. Additionally, these findings are consistent with those of previous studies that found that arousal and pleasure are higher at fast rather than slow tempo [71,82].

### 3.4. Discussion

As mentioned previously, social characteristics influence emotion evaluations. Considering the effects of SPL, musical emotions and tempo on emotional evaluation, the effects of social characteristics on communicating emotions under A Comme Amour (50 dBA), which did not have a significant emotional tendency and high musical evaluation, are discussed. Table 4 shows the average scores of musical evaluations, average d-values and the standard deviations of PAD emotional evaluations during communication with respect to social characteristics.

In terms of familiarity, the results of the ANOVA showed a significant effect on musical evaluations (df1 = 2, df2 = 49, F = 3.735, *p* < 0.05). Generally, musical evaluations were positive when participants’ familiarity was low (3.69) and medium (0.69), while it was negative (−0.1) in the high familiarity group. It was noticeable that the average scores of musical evaluations decreased from 3.69 to −0.1 as familiarity increased. As for PAD emotional evaluations, participants’ familiarity was not significantly correlated with pleasure (df1 = 2, df2 = 49, F = 1.112, *p* = 0.33, *ƞ^2^* = 0.043), arousal (df1 = 2, df2 = 49, F = 1.697, *p* = 0.19, *ƞ^2^* = 0.065) and dominance (df1 = 2, df2 = 49, F = 0.608, *p* = 0.55, *ƞ^2^* = 0.024). Other than pleasure that increased 0.88 in the low familiarity group, it decreased 0.06 and 0.45 in the middle and high familiarity groups, respectively. Both arousal and dominance increased in all familiarity degree groups; their d-values ranged from 0.19 to 1.88, and from 0.4 to 1.25, respectively. The findings indicate that during communication, the positive effect of music on communication was significant when participants were strangers; in contrast, when participants were highly familiar with each other, music tended to play a negative role in communication.

In terms of gender combination, the results of t-test showed that there was no significant effect of gender combinations on musical evaluation (t = −0.737, df = 50, *p* = 0.47, d = 0.20). The average score of musical evaluations in the mixed gender group was 1.77, which was much higher than the score in the single gender group (0.85). As for PAD emotional evaluations, there was no significant effect of gender combination on pleasure (t = −0.926, df = 50, *p* = 0.36, d = 0.26), arousal (t = −1.282, df = 50, *p* = 0.21, d = 0.36) and dominance (t = 0.623, df = 50, *p* = 0.54, d = 0.17). In general, pleasure decreased 0.27 in the single gender group, while increased 0.42 in the mixed gender group; both arousal and dominance increased in single and mixed gender groups. It was noticeable that there was a marked difference in gender groups when the musical evaluations were different. In the negative group, arousal decreased 0.67 in the single gender group and increased 1 in the mixed gender group; in the positive group, dominance increased 2.18 in the single gender group (mean = 2.18, SD = 1.888), which was 1.97 higher than in the mixed gender group (mean = 0.21, SD = 2.293). Further, in the positive group, the results of the t-test showed that gender combination was significantly correlated with dominance (t = 2.297, df = 23, *p* < 0.05). The findings indicate that, there was no significant effect of gender combination on musical and emotional evaluation during communication; when the musical evaluation was positive, dominance was much higher in the single gender groups.

In terms of the number of participants in the communication, the results of the t-test showed that there was no significant effect on musical evaluations (t = −1.049, df = 40.038, *p* = 0.300, d = 0.34). Generally, musical evaluations were positive in both one-on-one groups (1.03) and multi-participant groups (2.15). As for PAD emotional evaluations, the results of the t-test showed that the number of participants did not have a significant effect on pleasure (t = −1.568, df = 50, *p* = 0.123, d = 0.50), arousal (t = −0.792, df = 50, *p* = 0.432, d = 0.25), but was significant related with dominance (t = 1.709, df = 50, *p* < 0.05, d = 0.55). Pleasure decreased 0.26 in one-on-one groups while it increased 1.08 in multi-participant groups. Arousal and dominance both increased in one-on-one groups and multi-participant groups, from 1.00 to 1.69 and 0.54 to 1.85, respectively. The findings indicate that the number of participants did not have a significant effect on musical evaluation during communication, and compared to one-on-one group, a multi-paticipant group could enhance dominance.

## 4. Conclusions

Using objective measurements of the musical environment and a subjective questionnaire survey of musical evaluations and PAD emotional evaluations during communication, this study examined the different influences of various factors in the musical environment on communication emotions. Based on the results, several conclusions can be drawn.

First, SPL was significantly associated with musical evaluation of communication from 40 dBA to 60 dBA; overall, musical evaluation was negative when SPL was above 55 dBA. Arousal increased significantly with increases in musical SPL in the negative evaluation group. Therefore, the SPL of background music can appropriately be adjusted to below 50 dBA to maintain the positive effect of music on communication. Second, musical emotion had a significant influence on musical and emotional evaluations; specifically, musical evaluations were highest under joyful-sounding music. In general joyful- and stirring-sounding music could enhance pleasure and arousal efficiently. Therefore, generally, joyful-sounding music might be better for promoting communication. Third, musical tempo had a significant effect on musical evaluation and communicating emotion; faster music may enhance arousal and pleasure efficiently. In addition, social characteristics can affect communicating emotion to some extent. In terms of familiarity, the positive effect of music on communication was significant when participants were strangers. In terms of gender combination for the positive evaluation group, overall evaluations were higher in mixed-gender groups, and dominance was higher in single gender groups. As for the number of participants, compared to one-on-one group, multi-paticipant group had increased dominance.

The present study can help improve communicating emotions by regulating music in terms of SPL and musical emotion. The study had several limitations and certain questions need to be discussed further. First, the results of this study were mainly based on subjective evaluations and lacked data about physiological changes to support the above results. In further studies, physiological measurements can be used for more thorough assessments, such as, facial expression, skin conductance, and heart rate. Second, communication topic is one of the factors affecting emotion, but it was not considered in this study. Third, because of the limited size of an indoor space, communications that take place in outdoor space should be considered in future studies. Finally, participants in this study consisted of college students, and samples with a wider variety of characteristics, such as people of different ages, could be considered in further studies.

## Figures and Tables

**Figure 1 ijerph-17-02499-f001:**
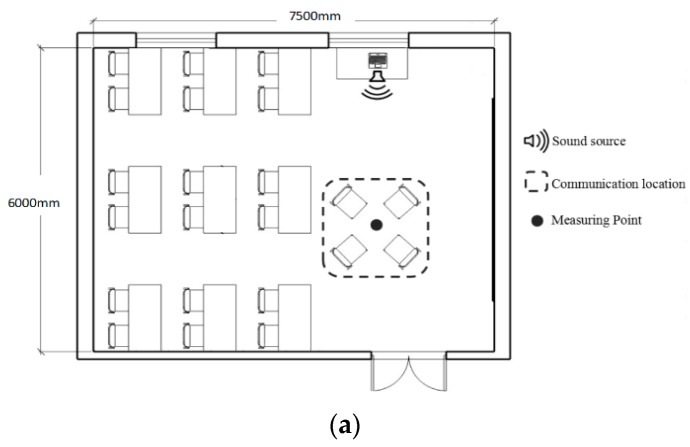
Indoor layout of the experimental room: (**a**) Plan of the experimental room (**b**) Selection of the experiment room.

**Figure 2 ijerph-17-02499-f002:**
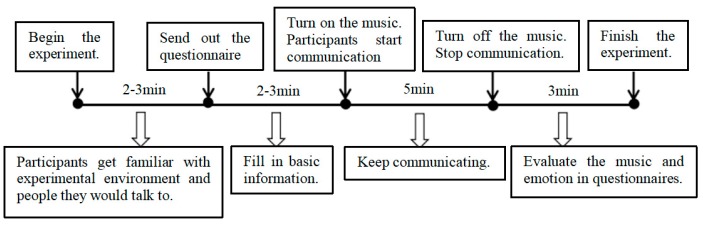
Experimental procedure showing the steps including familiarising experimental environment, listening to music to evaluating emotion.

**Figure 3 ijerph-17-02499-f003:**
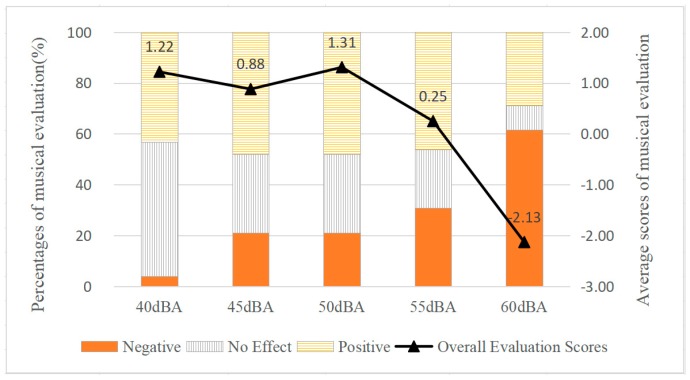
The Effect of SPL on Musical Evaluation during Communication.

**Figure 4 ijerph-17-02499-f004:**
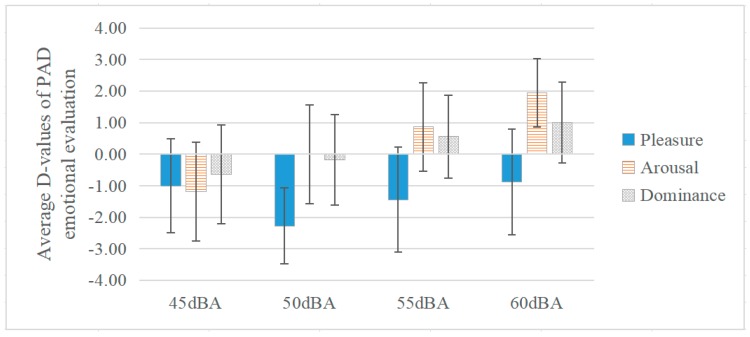
The Effect of SPL on PAD Emotional Evaluation during Communication (negative).

**Figure 5 ijerph-17-02499-f005:**
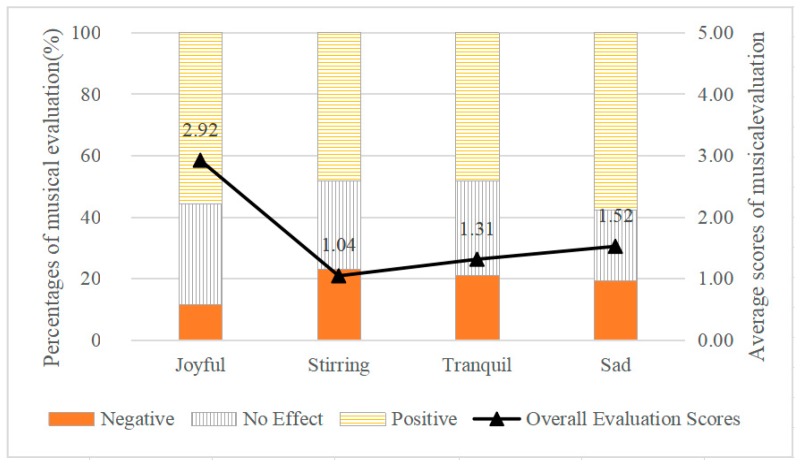
The Effect of Musical Emotions on Musical Evaluation during Communication.

**Figure 6 ijerph-17-02499-f006:**
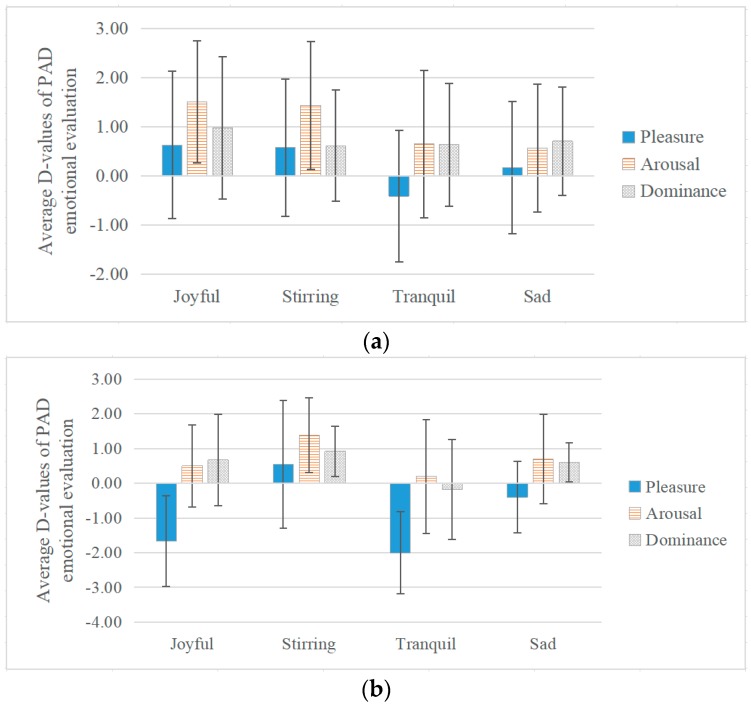
The Effect of Musical Emotion on PAD Emotional Evaluation during Communication: (**a**) In all participants (**b**) Negative group.

**Figure 7 ijerph-17-02499-f007:**
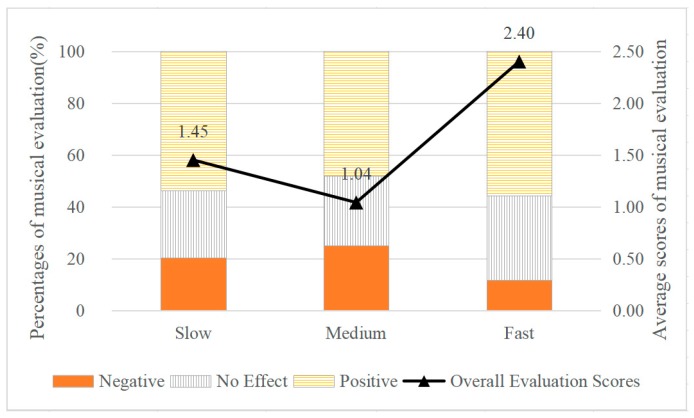
The Effect of Musical Tempo on Musical Evaluation during Communication.

**Figure 8 ijerph-17-02499-f008:**
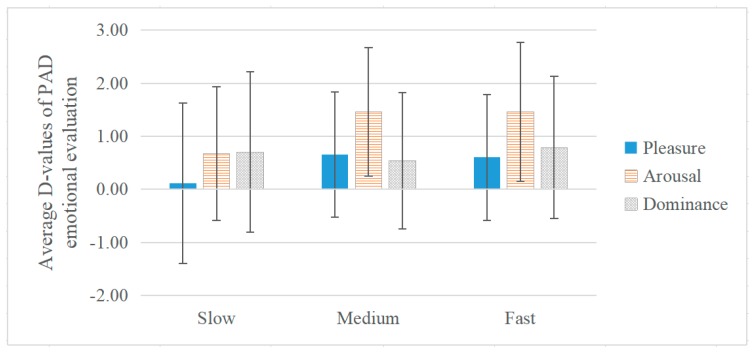
The Effect of Tempo on PAD Emotional Evaluation during Communication.

**Table 1 ijerph-17-02499-t001:** Social Characteristics of Participants.

Social Characteristics		Number
Gender combination	Single gender	26
	Mixed gender	26
Familiarity	Low degree	16
	Middle degree	16
	High degree	20
Number of Participants	One-on-one group (2)	38
	Multi-participants group (3/4)	14

**Table 2 ijerph-17-02499-t002:** Subjective evaluation descriptions during communication.

Subjective Evaluation		Description
Basic Information		Name, Gender, Age
Overall Musical Evaluation	No effect (0)	Music has no effect
	Conducive (1 to 10)	A little conducive to Conducive very much
	Distractive (–1 to –10)	A little distractive to Distractive very much
Emotion Dimension	Pleasure (1 to 10)	Depressed to Satisfied, Unhappy to Happy, Restless to Comfortable, Angry to Glad
	Arousal (1 to 10)	Peaceful to Fevered, Unexcited to Excited, Relaxed to Stimulated, Drowsy to Awakened
	Dominance (1 to 10)	Passive to Active, Controlled to Uncontrolled

**Table 3 ijerph-17-02499-t003:** Typical Musical Emotion Evaluation Survey.

Evaluation	Music Title	Joyful	Agitated	Tranquil	Sad
Joyful	Dance of The Golden Snake	21 (70%)	9 (30%)	0 (0%)	0 (0%)
	Ode an die Freude	22 (73.33%)	3 (10%)	5 (16.67%)	0 (0%)
	Spring Festival Prelude	25(83.33%)	3 (10%)	0 (0%)	2 (6.67%)
Stirring	Croatian Rhapsody	6 (20%)	23 (76.67%)	0 (0%)	1 (3.33%)
	Carmen Overture	14 (46.67%)	13 (43.33%)	1 (3.33%)	2 (6.67%)
	Athletes March	5 (16.67%)	24 (80%)	1 (3.33%)	0 (0%)
Tranquil	A Comme Amour	2 (6.67%)	0 (0%)	24 (80%)	4 (13.33%)
	The Blue Danube	6 (20%)	5 (16.67%)	14 (46.67%)	5 (16.67%)
	For Elise	6 (20%)	0(0%)	21 (70%)	3 (10%)
Sad	River Water	0 (0%)	7 (23.33%)	1 (3.33%)	22 (73.33%)
	MARIAGE D’AMOUR	12 (40%)	0 (0%)	14 (46.67%)	4 (13.33%)
	Butterfly Lovers	4 (13.33%)	2 (6.67%)	0 (0%)	27 (90%)

**Table 4 ijerph-17-02499-t004:** Musical and Emotional Evaluations in terms of Social Characteristics.

Social Characteristics		Musical Evaluation		D-values of Emotional Evaluation					
ANOVA				Pleasure		Arousal		Dominance	
Familiarity	Low degree	**3.69**		0.88		1.88		1.06	
	Middle degree	0.69		−0.06		0.19		1.25	
	High degree	-0.1		−0.45		1.84		0.40	
**T-test**		**Mean**	**SD**	**Mean**	**SD**	**Mean**	**SD**	**Mean**	**SD**
Gender Combination	Single gender	0.85	4.628	−0.27	2.616	0.69	3.222	1.08	2.365
	Mixed gender	1.77	4.403	0.42	2.774	1.65	2.058	0.65	2.529
Number of Participants	2	1.03	4.966	−0.26	2.613	1.00	2.956	0.54	2.292
	2–4	2.15	2.609	1.08	2.783	1.69	1.843	1.85	2.672

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
