# Peer review of "Effects of the Musical Sound Environment on Communicating Emotion"

_ijerph, 2020, doi:10.3390/ijerph17072499_

Round 1
Reviewer 1 Report
I would like to encourage these authors to continue in their research on music and its effects in the built environment. The authors have done extensive classification and measurement of their environment in terms of light and sound pressure levels, etc. I strongly suggest that the authors, who are well-versed in environmental measurements, collaborate with a music psychologist to meaningfully contribute to this field.
Unfortunately, this paper is not ready for publication for three reasons:
The paper has extensive technical errors, such as the misclassifying of music excerpts as fast that are actually slow. The paper demonstrates insufficient background knowledge. The authors seem unaware of the entire soundscape literature, which is essential when thinking about music in the built environment. The paper has very little novelty. The authors' main measurement is a simple subjective PAD measurement of emotions; furthermore the sample size of participants is small. Much more sophisticated phjysiological measurements have been made long ago (Krumhansl, C. L. (1997). An exploratory study of musical emotions and psychophysiology. Canadian Journal of Experimental Psychology/Revue canadienne de psychologie expérimentale, 51(4), 336.). The current study's authors' basic measurements of subjective emotions in musical excerpts were made in the 1930s by Kate Hevner: Hevner, K. (1936). Experimental studies of the elements of expression in music. The American Journal of Psychology, 48(2), 246-268. Finally, there is a problem with the English language. While I think the authors have done well generally with their English, the most important phrase in the title, "communication emotion", is not grammatical English, and is thus ambiguous.ESSENTIAL ISSUES
Line 3: A major problem with this paper is the meaning of the phrase "communication emotion", which is in the title and is the central idea of the paper. I am a native English speaker, and this phrase is not grammatical English, and it is not a technical term that I am aware of in the music psychology literature. Do the authors mean "communication of emotion", "communicating emotion" or "communication in the environment of emotion"? Because the authors mention "emotional contagion", I presume they mean "communicating emotion", but I am not sure…
Line 64: The authors make a totally false claim "However, there is a lack of research on positive effects of the acoustic environment on emotion." There is a large and long-standing literature on the positive effects of both environmental sounds and of music upon the environment and its effects on emotions. The field is called soundscape. The authors can start by reading this: Yang, W., & Kang, J. (2005). Soundscape and sound preferences in urban squares: a case study in Sheffield. Journal of Urban Design, 10(1), 61-80.
Line 187: the authors cite three sources that are not listed in the reference section: (Mirella and Foxall, 2006; Christian and Wachsmuth, 2010; Mehrabian 188 and Russell, 1974). These references were also incorrectly cited, as the authors' names (Mirella Yani-de-Soriano and Christian Becker-Asano) were listed by their given names rather than by their family names.
Line 234: the authors state that their analysis of tempo using MixMeister BPM Analyzer determined that the tempo of A Comme Amour was 137.03 bpm, which is plainly double the correct value. A Comme Amour is a calm piece, the tempo is scored as "andante" where quarter notes are listed at 60-70 bpm. This kind of major error would mean that the author's analysis of tempo would be completely meaningless.
DETAILS
Line 12: Add "the": "Acoustic environment" à "The acoustic environment"
Line 15: change "communication" to communicating": "the effects of music on communication emotion" à "the effects of music on communicating emotion"
Line 37: change "leading" to "lead": "may leading to heart disease" à "may lead to heart disease"
Line 65: add "the": "of research on positive effects of the acoustic environment on emotion" à "of research on the positive effects of the acoustic environment on emotion"
Linre 87: change "communication" to communicating": "effects of music on communication emotion have been" à "effects of music on communicating emotion have been"
Line 157-8: change both instances of "was" to "changed": "When the number of participants in communication was from 2 to 4, the RT of the room was 158 from 0.59 s to 0.6 s" à "When the number of participants in communication changed from 2 to 4, the RT of the room changed 158 from 0.59 s to 0.6 s"
Author Response
Effects of Musical Sound Environment on Communicating Emotion
Qi Meng, Jiani Jiang, Fangfang Liu, XiaoduoXu
We greatly appreciate the further comments/suggestions given by the reviewer(s). Modifications have been made accordingly – all comments have been taken, as described below.
I would like to encourage these authors to continue in their research on music and its effects in the built environment. The authors have done extensive classification and measurement of their environment in terms of light and sound pressure levels, etc. I strongly suggest that the authors, who are well-versed in environmental measurements, collaborate with a music psychologist to meaningfully contribute to this field.
Unfortunately, this paper is not ready for publication for three reasons:
The paper has extensive technical errors, such as the misclassifying of music excerpts as fast that are actually slow. The paper demonstrates insufficient background knowledge. The authors seem unaware of the entire soundscape literature, which is essential when thinking about music in the built environment. The paper has very little novelty. The authors' main measurement is a simple subjective PAD measurement of emotions; furthermore the sample size of participants is small. Much more sophisticated phjysiological measurements have been made long ago (Krumhansl, C. L. (1997). An exploratory study of musical emotions and psychophysiology. Canadian Journal of Experimental Psychology/Revue canadienne de psychologie expérimentale, 51(4), 336.). The current study's authors' basic measurements of subjective emotions in musical excerpts were made in the 1930s by Kate Hevner: Hevner, K. (1936). Experimental studies of the elements of expression in music. The American Journal of Psychology, 48(2), 246-268. Finally, there is a problem with the English language. While I think the authors have done well generally with their English, the most important phrase in the title, "communication emotion", is not grammatical English, and is thus ambiguous.
In order to present the research questions clearly, the following revisions were made in the INTRODUCTION, Lines 89-99, as follows:
“In terms of musical emotion, a significant body of research has found that the relationships between the emotion perceived and the emotion felt in music do not always match. In general, music can arouse emotions similar to the emotional quality perceived in music, whereas sometimes, fearful music is perceived as negative but felt as positive (Gabrielsson, 2002; Garrido and Schubert, 2011). Besides, the emotion felt are frequently rated same as or lower than the emotion perceived, which may be attributed to inhibition of emotion contagion (Schubert, 2013). Studies have mainly focused on how listeners evaluate musical emotions when listening to music on purpose (Krumhansl and Carol, 1997; Hevner, 1936). However, an individual listens to music in isolation and, even when listening occurs in a social setting, it may not co-occur with social interactions; the combined effects of music and social interaction on communicating emotions have been less studied.”
The following references are also added:
Hevner K. Experimental studies of the elements of expression in music. American J Psychology, 1936, 48(2):246-268.
Garrido, S., & Schubert, E. (2011). Individual differences in the enjoyment of negative emotion in music: A literature review and experiment. Music Perception: An Interdiscliplinary Journal, 28, 279–2964. doi:10.1525/mp.2011.28.3.279
Schubert, E. (2013). Emotion felt by the listener and expressed by the music: Literature review and theoretical perspectives. Frontiers in Psychology, 4. doi:10.3389/fpsyg.2013.00837
Krumhansl, and Carol, L. . (1997). An exploratory study of musical emotions and psychophysiology. Canadian Journal of Experimental Psychology/Revue Canadienne De Psychologie Expérimentale, 51(4), 336-353.
ESSENTIAL ISSUES
Line 3: A major problem with this paper is the meaning of the phrase "communication emotion", which is in the title and is the central idea of the paper. I am a native English speaker, and this phrase is not grammatical English, and it is not a technical term that I am aware of in the music psychology literature. Do the authors mean "communication of emotion", "communicating emotion" or "communication in the environment of emotion"? Because the authors mention "emotional contagion", I presume they mean "communicating emotion", but I am not sure…
Thank you for your suggestion. In the revised version, all instances of “communication emotion” has have been revised to “communicating emotion”.
Line 64: The authors make a totally false claim "However, there is a lack of research on positive effects of the acoustic environment on emotion." There is a large and long-standing literature on the positive effects of both environmental sounds and of music upon the environment and its effects on emotions. The field is called soundscape. The authors can start by reading this: Yang, W., & Kang, J. (2005). Soundscape and sound preferences in urban squares: a case study in Sheffield. Journal of Urban Design, 10(1), 61-80.
In the INTRODUCTION of the revised manuscript, information about soundscape literature has been added. Lines 58-67:
“As for the acoustic perception, the term ‘soundscape’ was coined by Schafer, who defined it as a sonic environment, with an emphasis on the way it is perceived and understood by individuals or society (Brown et al., 2011). In 2014, the International Organization of Standardization developed a broader definition of soundscape: acoustic environment as perceived or experienced and/or understood by a person (ISO, 2014). Assessment of soundscape is part of sensory aesthetics research, and the aesthetic response of surroundings is considered to be a mix of high pleasure, excitement, and relaxation (Lang, 1988; Nasar, 1989). In terms of positive effects, a study suggested that nature- and culture-related sounds, such as fountains, birds singing, bells, and music from clock, which induce tranquil and pleasant feelings, are preferred in urban squares, as opposed to artificial sounds (Yang and Kang, 2005)”
The following references are also added:
Brown, A.L., Kang, J., Gjestland, T., 2011. Towards standardization in soundscape preference assessment. Appl. Acoust. 72 (6), 387–392.
ISO 12913-1, 2014. Acoustics–Soundscape–Part 1: Definition and Conceptual Framework.
Lang, J. (1988) Symbolic aesthetics in architecture: toward a research agenda, in: J. L. Nasar (Ed.)Environmental Aesthetics (Cambridge: Cambridge University Press).
Nasar, J. L. (1989) Perception, cognition, and evaluation of urban places, in: I. Altman & E. H. Zube(Eds) Public Places and Spaces (New York: Plenum Press).
Yang, Wei , and J. Kang . "Soundscape and Sound Preferences in Urban Squares: A Case Study in Sheffield." Journal of Urban Design 10.1(2005):61-80.
Line 187: the authors cite three sources that are not listed in the reference section: (Mirella and Foxall, 2006; Christian and Wachsmuth, 2010; Mehrabian 188 and Russell, 1974). These references were also incorrectly cited, as the authors' names (Mirella Yani-de-Soriano and Christian Becker-Asano) were listed by their given names rather than by their family names.
This has been corrected in the revised version in section 2.3 as well as the references, as follows:
“(Mehrabian and Russell, 1974; Yani-De-Soriano and Foxall, 2006; and Becker-Asano and Wachsmuth, 2010;)”
The following references have been corrected:
Becker-Asano, C., Wachsmuth, I., 2010. Affective computing with primary and secondary emotions in a virtual human. Auton. Agent. Multi-Agent Sys. 20, 32–49. https://doi.org/10.1007/s10458-009-9094-9.”
Mehrabian, A. , & Russell, J. A. . (1974). The basic emotional impact of environments. Perceptual and Motor Skills, 38(1), 283-301.”
Yani-De-Soriano, M.M., Foxall, G.R., 2006. The emotional power of place: The fall and rise of dominance in retail research. Journal of Retailing and Consumer Services13, 403–416. https://doi.org/10.1016/j.jretconser.2006.02.007.
Line 234: the authors state that their analysis of tempo using MixMeister BPM Analyzer determined that the tempo of A Comme Amour was 137.03 bpm, which is plainly double the correct value. A Comme Amour is a calm piece, the tempo is scored as "andante" where quarter notes are listed at 60-70 bpm. This kind of major error would mean that the author's analysis of tempo would be completely meaningless.
We apologize for these mistakes. Information about the tempo of the music excerpts has been revised in section 2.4.2, Lines 259-261, as follows:
“A metronome was used to analyse the tempo of the four music excerpts selected in 2.4.1. The music excerpts had the following BPM values: Spring Festival Prelude was 144 (allegro), Athletes March was 110 (allegretto), A Comme Amour was 70 (andante), and Butterfly Lovers was 50 (adagio).”
DETAILS
Line 12: Add "the": "Acoustic environment" à "The acoustic environment"
This has been corrected in revised version.
Line 15: change "communication" to communicating": "the effects of music on communicationemotion" à "the effects of music on communicating emotion"
This has been corrected in revised version.
Line 37: change "leading" to "lead": "may leading to heart disease" à "may lead to heart disease"
This has been corrected in revised version.
Line 65: add "the": "of research on positive effects of the acoustic environment on emotion" à "of research on the positive effects of the acoustic environment on emotion"
This has been corrected in revised version.
Linre 87: change "communication" to communicating": "effects of music on communicationemotion have been" à "effects of music on communicating emotion have been"
This has been corrected in revised version.
Line 157-8: change both instances of "was" to "changed": "When the number of participants in communication was from 2 to 4, the RT of the room was 158 from 0.59 s to 0.6 s" à "When the number of participants in communication changed from 2 to 4, the RT of the room changed 158 from 0.59 s to 0.6 s"
This has been corrected in revised version.
Reviewer 2 Report
In the present paper, the authors present results from their study on the effect of background music on communication. While the research question is interesting and novel to my knowledge, it is not possible to evaluate the study they performed. The report lacks clarity in all sections. For instance, I am not sure at all what the authors mean by “emotion communication” in their survey. Were participants asked to rate how they felt after the conversation? Were they asked to rate how they felt about their partner? Were they asked to rate how they felt the conversation went? It is not clear at all, so all subsequent reported results are entirely uninterpretable.
In addition, the reported methods and results do not demonstrate grasp of inferential statistics. The authors report a power analysis, which doesn’t seem to map clearly to any of the statistical tests they subsequently ran. Perhaps most curiously, they specified an alpha level in their power analysis, but then did not use any p-values in reported results, and were missing degrees of freedom for the reported F-tests.
The manuscript would additionally likely benefit from an editor with professional proficiency in English, as the wording and grammatical structure is confusing at some points.
Overall, while I feel the described research question is noteworthy, I cannot recommend the manuscript for publication given these issues, as well as the apparent holes in their reporting of the music and emotion literature. See further comments below.
MAJOR ISSUES
Ln 77-88
The paper would benefit from a more thorough discussion about the distinction between felt and perceived emotion in music. While some work is mentioned that is tangential to this (for example, effects on consumer behavior) and one paper mentioned that compares felt and perceived emotion (Hunter & Schellenberg, 2010), there is a large body of research on this topic that is relevant.
Schubert, E. (2013). Emotion felt by the listener and expressed by the music: Literature review and theoretical perspectives. Frontiers in Psychology, 4. doi:10.3389/fpsyg.2013.00837
Ln 93-95: “In terms of age, compared to older listeners, younger children had more difficulty identifying sad- or peaceful-sounding excerpts correctly, but adult-like accuracy was reached by 11 years of age (Hunter et al., 2008).”
This is another good example in which the distinction between felt and perceived emotions is quite unclear. The paragraph begins “how we respond to music,” but these studies are about the evaluation of musical emotion instead of the child’s response. The cited literature about music education, as well, is about perceived emotion. I think it is reasonable to include this information but the distinction and relevance needs to be clearer.
Participants, Ln. 161: “To ensure adequate statistical power, G*Power was used to analyse the minimum sample size of subjects, assuming an effect size of d = 0.5, α err prob = 0.05, Power (1-β err prob = 0.8). For the main research questions, minimum sample size was 27 in paired-samples t-test and was 40 in One-way ANOVA (within-group). Therefore, considering the variety of gender composition, subjects’ familiarity, and grade of students, 52 participants were randomly sampled from 500 college students.”
What are the “main research questions” (i.e. the specific statistical tests) described for the paired-sample test and the one-way ANOVA mentioned? The authors seem to be suggesting that they tested more participants than the minimum recommended sample because they wanted to account for the additional factors of gender, familiarity, and age. However, for each additional factor, many times more participants would be required.
Methods
Design is very unclear. The power analysis was done on paired tests, but Fig. 2 suggests that each participant only gets one level of the possible conditions (music emotion, SPL, and tempo). Also, what are the hypotheses of the study?
Results
Results are additionally not satisfactorily reported. It is very odd that a power analysis was reported, and alpha is controlled in the G*Power analysis, but no p-values are reported. Since the alpha is intended to control the p-values, what is the purpose of the power analysis without p-values? Simply reporting effect size is not sufficient – both are important for interpretation. It is also quite confusing that multiple measures of effect size (both d and partial eta squared) are reported, what is the reason for this?
Also, degrees of freedom and p values are missing for all F-tests.
Questionnaire survey
In the survey “positive” and “negative” are used very differently than usual in the music literature. Positive and negative refer to valence (happy/joy/peaceful vs. sad/angry/upset), rather than level of distraction, which seems to be how it was used here. Especially in the context of emotion, using positive/negative seems to be very misleading.
Ln 221 “It is widely accepted that the emotion perceived by listeners is in agreement with the emotional expression in the music (Gabrielsson, 2002)” While this is often true, it is certainly not necessarily true. This is an especially important distinction in the case of “sad” music, which people often report as pleasurable to listen to.
Garrido, S., & Schubert, E. (2011). Individual differences in the
enjoyment of negative emotion in music: A literature review
and experiment. Music Perception: An Interdiscliplinary
Journal, 28, 279–2964. doi:10.1525/mp.2011.28.3.279
Huron, D. (2011). Why is sad music pleasurable? A possible role
for prolactin. Musicae Scientiae, 15, 146–158. doi:10.1177/
1029865911401171
MINOR ISSUES
Ln 58: “In terms of traffic noise, exposure to traffic noise can increase annoyance, and even cause hypertension and cardiovascular disease (Barregard, 2011).”
Though I am unfamiliar with the study above, it seems unlikely to me that this paper showed a causal relationship. Please address if they simply showed an association.
Ln 90: “previous studies have found that social characteristics, such as gender, age, familiarity, and musical education”
Familiarity with the music is not really a social characteristic. This makes it seem like you mean social familiarity with another individual, which is quite confusing.
Table 1
This seems like it belongs in supplementary material rather than the main body of the paper.
Ln 234 “In 234 terms of tempo, there were two groups: a low BPM group (medium, average BPM value = 108.66) 235 and a high BPM group (fast, average BPM value = 137).”
What does “low BPM group (medium)” mean? This is contradictory.
Author Response
In the present paper, the authors present results from their study on the effect of background music on communication. While the research question is interesting and novel to my knowledge, it is not possible to evaluate the study they performed. The report lacks clarity in all sections. For instance, I am not sure at all what the authors mean by “emotion communication” in their survey. Were participants asked to rate how they felt after the conversation? Were they asked to rate how they felt about their partner? Were they asked to rate how they felt the conversation went? It is not clear at all, so all subsequent reported results are entirely uninterpretable.
We have clarified this in the revised version in Lines 114-115, as follows:
“Therefore, this study aimed to investigate the effects of music on communicating emotion by rating how participants felt before and after the conversation based on the following research questions....”
In addition, the reported methods and results do not demonstrate grasp of inferential statistics. The authors report a power analysis, which doesn’t seem to map clearly to any of the statistical tests they subsequently ran. Perhaps most curiously, they specified an alpha level in their power analysis, but then did not use any p-values in reported results, and were missing degrees of freedom for the reported F-tests.
Thank you for your suggestion. The degrees of freedom and p-values have been added in the RESULTS.
The manuscript would additionally likely benefit from an editor with professional proficiency in English, as the wording and grammatical structure is confusing at some points.
We agree with you. The revised manuscript has been proofread and checked for grammar, syntax, and phrases by the Elsevier language editing service.
MAJOR ISSUES
Ln 77-88
The paper would benefit from a more thorough discussion about the distinction between felt and perceived emotion in music. While some work is mentioned that is tangential to this (for example, effects on consumer behavior) and one paper mentioned that compares felt and perceived emotion (Hunter & Schellenberg, 2010), there is a large body of research on this topic that is relevant.
Schubert, E. (2013). Emotion felt by the listener and expressed by the music: Literature review and theoretical perspectives. Frontiers in Psychology, 4. doi:10.3389/fpsyg.2013.00837
In the revised version, sentences not related to the topic of this study have been deleted, and a detailed discussion about the difference between felt and perceived emotion in music has been added in Lines 90-96, as follows:
“In terms of musical emotion, a significant body of research has found that the relationships between the emotion perceived and the emotion felt in music do not always match. In general, music can arouse emotions similar to the emotional quality perceived in music, whereas sometimes, fearful music is perceived as negative but felt as positive (Gabrielsson, 2002; Garrido and Schubert, 2011). Besides, the emotion felt are frequently rated same as or lower than the emotion perceived, which may be attributed to inhibition of emotion contagion (Schubert, 2013).”
The following references have also been added:
Garrido, S., & Schubert, E. (2011). Individual differences in the enjoyment of negative emotion in music: A literature review and experiment. Music Perception: An Interdiscliplinary Journal, 28, 279–2964. doi:10.1525/mp.2011.28.3.279
Schubert, E. (2013). Emotion felt by the listener and expressed by the music: Literature review and theoretical perspectives. Frontiers in Psychology, 4. doi:10.3389/fpsyg.2013.00837
Ln 93-95: “In terms of age, compared to older listeners, younger children had more difficulty identifying sad- or peaceful-sounding excerpts correctly, but adult-like accuracy was reached by 11 years of age (Hunter et al., 2008).”
This is another good example in which the distinction between felt and perceived emotions is quite unclear. The paragraph begins “how we respond to music,” but these studies are about the evaluation of musical emotion instead of the child’s response. The cited literature about music education, as well, is about perceived emotion. I think it is reasonable to include this information but the distinction and relevance needs to be clearer.
This has been clarified in the INTRODUCTION section in Lines 104-107, as follows:
“In terms of age, compared to younger children, older listeners can recognize the perceived emotion in music more correctly. Besides, older listeners hold more emotional experiences, and music sometimes would revoke their personal and private memories, which may also lead to different felt emotions (Hunter et al., 2008; Schubert, 2013).”
The following reference has also been added:
Schubert, E. (2013). Emotion felt by the listener and expressed by the music: Literature review and theoretical perspectives. Frontiers in Psychology, 4. doi:10.3389/fpsyg.2013.00837
Participants, Ln. 161: “To ensure adequate statistical power, G*Power was used to analyse the minimum sample size of subjects, assuming an effect size of d = 0.5, α err prob = 0.05, Power (1-β err prob = 0.8). For the main research questions, minimum sample size was 27 in paired-samples t-test and was 40 in One-way ANOVA (within-group). Therefore, considering the variety of gender composition, subjects’ familiarity, and grade of students, 52 participants were randomly sampled from 500 college students.”
What are the “main research questions” (i.e. the specific statistical tests) described for the paired-sample test and the one-way ANOVA mentioned?
This has been added in Lines 175-176, as follows:
“For the main research questions regarding the effects of SPL, musical emotions, and tempo on communicating emotions, the minimum required sample size was 40 in the one-way ANOVA (within-group).”
The specific statistical tests are added in section 2.6, Line 318-321, as follows:
“...Analysis of variance (ANOVA) was used to test for significant differences under different levels of background SPL, musical emotion, tempo (within-group), and familiarity with the participants (between-group). Independent-sample t-tests were performed to assess gender differences, and the difference between one-on-one and multi-participant groups....”
The authors seem to be suggesting that they tested more participants than the minimum recommended sample because they wanted to account for the additional factors of gender, familiarity, and age. However, for each additional factor, many times more participants would be required.
In the revised version, this has been changed in section 2.2, Lines 172-176, as follows :
“To ensure adequate statistical power, G*Power was used to analyse the minimum sample size of subjects, assuming an effect size of d = 0.5, α err prob = 0.05, Power (1-β err prob = 0.8). For the main research questions regarding the effects of SPL, musical emotions, and tempo on communicating emotions, the minimum required sample size was 40 in the one-way ANOVA (within-group).”
Methods
Design is very unclear. The power analysis was done on paired tests, but Fig. 2 suggests that each participant only gets one level of the possible conditions (music emotion, SPL, and tempo). Also, what are the hypotheses of the study?
In the revised version, paired test has been changed to ANOVA in section 2.6, Lines 320-323, as follows:
“...Analysis of variance (ANOVA) was used to test for significant differences under different levels of background SPL, musical emotion, tempo (within-group), and familiarity with the participants (between-group). Independent-sample t-tests were performed to assess gender differences, and the difference between one-on-one and multi-participant groups....”
Results
Results are additionally not satisfactorily reported. It is very odd that a power analysis was reported, and alpha is controlled in the G*Power analysis, but no p-values are reported. Since the alpha is intended to control the p-values, what is the purpose of the power analysis without p-values? Simply reporting effect size is not sufficient – both are important for interpretation. It is also quite confusing that multiple measures of effect size (both d and partial eta squared) are reported, what is the reason for this?
Also, degrees of freedom and p values are missing for all F-tests.
Thank you for your suggestion, the degrees of freedom and p-values have been added in the RESULTS.
Questionnaire survey
In the survey “positive” and “negative” are used very differently than usual in the music literature. Positive and negative refer to valence (happy/joy/peaceful vs. sad/angry/upset), rather than level of distraction, which seems to be how it was used here. Especially in the context of emotion, using positive/negative seems to be very misleading.
In order to avoid the misunderstanding of “positive” and “negative”, in the revised version, “positive” has been changed to “conductive”, and “negative” has been changed to “distractive” in Table 2.
Ln 221 “It is widely accepted that the emotion perceived by listeners is in agreement with the emotional expression in the music (Gabrielsson, 2002)” While this is often true, it is certainly not necessarily true. This is an especially important distinction in the case of “sad” music, which people often report as pleasurable to listen to.
Garrido, S., & Schubert, E. (2011). Individual differences in the
enjoyment of negative emotion in music: A literature review
and experiment. Music Perception: An Interdiscliplinary
Journal, 28, 279–2964. doi:10.1525/mp.2011.28.3.279
Huron, D. (2011). Why is sad music pleasurable? A possible role
for prolactin. Musicae Scientiae, 15, 146–158. doi:10.1177/
1029865911401171
In the revised version, this sentence has been deleted, and the discussion about the unmatched emotion has added in INTRODUCTION, Lines 79-82 and Lines 89-95, as follows:
“...Evidence against a strict cognitivist position suggests that music can induce some sort of an emotional response (Hunter and Schellenberg, 2010; Krumhansl and Carol, 1997). There are nine common emotions (wonder, transcendence, tenderness, sadness, nostalgia, peacefulness, power, joyful activation, and tension) evoked by music (Huron, 2011).”
“...In terms of musical emotion, a significant body of research has found that the relationships between the emotion perceived and the emotion felt in music do not always match. In general, music can arouse emotions similar to the emotional quality perceived in music, whereas sometimes, fearful music is perceived as negative but felt as positive (Gabrielsson, 2002; Garrido and Schubert, 2011). Besides, the emotion felt are frequently rated same as or lower than the emotion perceived, which may be attributed to inhibition of emotion contagion (Schubert, 2013).”
The following references are also added:
Garrido, S., & Schubert, E. (2011). Individual differences in the enjoyment of negative emotion in music: A literature review and experiment. Music Perception: An Interdiscliplinary Journal, 28, 279–2964. doi:10.1525/mp.2011.28.3.279
Huron, D. (2011). Why is sad music pleasurable? A possible role for prolactin. Musicae Scientiae, 15, 146–158. doi:10.1177/1029865911401171
Krumhansl, and Carol, L. . (1997). An exploratory study of musical emotions and psychophysiology. Canadian Journal of Experimental Psychology/Revue Canadienne De Psychologie Expérimentale, 51(4), 336-353.
Schubert, E. (2013). Emotion felt by the listener and expressed by the music: Literature review and theoretical perspectives. Frontiers in Psychology, 4. doi:10.3389/fpsyg.2013.00837
MINOR ISSUES
Ln 58: “In terms of traffic noise, exposure to traffic noise can increase annoyance, and even cause hypertension and cardiovascular disease (Barregard, 2011).”
Though I am unfamiliar with the study above, it seems unlikely to me that this paper showed a causal relationship. Please address if they simply showed an association.
In the revised version, this sentence has been deleted.
Ln 90: “previous studies have found that social characteristics, such as gender, age, familiarity, and musical education”
Familiarity with the music is not really a social characteristic. This makes it seem like you mean social familiarity with another individual, which is quite confusing.
Thank you for your suggestion. In in the revised version, “familiarity” has been deleted.
Table 1
This seems like it belongs in supplementary material rather than the main body of the paper.
In the revised version, Table 1 has been deleted.
Ln 234 “In 234 terms of tempo, there were two groups: a low BPM group (medium, average BPM value = 108.66) 235 and a high BPM group (fast, average BPM value = 137).”
What does “low BPM group (medium)” mean? This is contradictory.
In the revised version, this has been changed to slow, medium, and fast in section 2.4.2, Line 259-266, as follows:
“...The music excerpts had the following BPM values: Spring Festival Prelude was 144 (allegro), Athletes March was 110 (allegretto), A Comme Amour was 70 (andante), and Butterfly Lovers was 50 (adagio). Additionally, the melodic structures of the chosen music excerpts are generally regular, with little tempo fluctuations; therefore, the average BPM could be used in tempo analysis. In the following sections, the tempo is divided into three groups for comparison, namely slow tempo (A Comme Amour, and Butterfly Lovers), medium tempo (Athletes March), and fast tempo (Spring Festival Prelude).
Reviewer 3 Report
This paper aims to study the effect of music on communication emotions in terms of sound pressure level (SPL), music emotion and tempo, and draws corresponding conclusions. The work of this paper is practical and logical. However, there are some problems to be further improved.
Considering the diversity of musical emotions, this paper chooses four types of music: joyful, stirring, tranquil and sad. The author did not explain in the paper how many specific music emotions were divided into, and then chose these four. Is there a theoretical basis for this classification? The four musical emotions selected in the paper can’t cover all musical emotions. Regarding the classification of musical emotions, please refer to the following papers:
[1] Hevner K. Experimental studies of the elements of expression in music. American J Psychology, 1936, 48(2):246-268.
[2] Zentner, M, D. Grandjean, and K. R. Scherer. Emotions evoked by the sound of music: characterization, classification, and measurement. Emotion 8.4(2008):494-521.
[3] Siqi Huang, Li Zhou, Zhentao Liu, Shan Ni, Jingxian He. Empirical Research on a Fuzzy Model of Music Emotion Classification based on Pleasure-Arousal Model. The 37th Chinese Control Conference, pp. 3239-3244, Wuhan, China, July 25-27, 2018.
The selection of experimental materials in the tempo part of the paper is based on the average BPM value of the selected music clips. Does it take into account the tempo fluctuations in the music? For example: the average BPM value of music in the first 15 seconds is 30, the average BPM value of music in the last 15 seconds is 170, and the average BPM value of the entire song is 100.
The paper only studies the influence of medium tempo (average BPM value = 108.66) and fast (average BPM value = 137) on the emotion of communication, and draws the conclusion that the tempo of music has little effect on emotion. This study lacks the slow tempo. The empirical study of the effects of communication emotions is not comprehensive enough and the conclusions are one-sided. Consider adding experimental music samples for additional research.
Author Response
Effects of Musical Sound Environment on Communicating Emotion
Qi Meng, Jiani Jiang, Fangfang Liu, XiaoduoXu
We greatly appreciate the further comments/suggestions given by the reviewer(s). Modifications have been made accordingly – all comments have been taken, as described below.
This paper aims to study the effect of music on communication emotions in terms of sound pressure level (SPL), music emotion and tempo, and draws corresponding conclusions. The work of this paper is practical and logical. However, there are some problems to be further improved.
Considering the diversity of musical emotions, this paper chooses four types of music: joyful, stirring, tranquil and sad. The author did not explain in the paper how many specific music emotions were divided into, and then chose these four. Is there a theoretical basis for this classification? The four musical emotions selected in the paper can’t cover all musical emotions. Regarding the classification of musical emotions, please refer to the following papers:
[1] Hevner K. Experimental studies of the elements of expression in music. American J Psychology, 1936, 48(2):246-268.
[2] Zentner, M, D. Grandjean, and K. R. Scherer. Emotions evoked by the sound of music: characterization, classification, and measurement. Emotion 8.4(2008):494-521.
[3] Siqi Huang, Li Zhou, Zhentao Liu, Shan Ni, Jingxian He. Empirical Research on a Fuzzy Model of Music Emotion Classification based on Pleasure-Arousal Model. The 37th Chinese Control Conference, pp. 3239-3244, Wuhan, China, July 25-27, 2018.
Thank you for your suggestions. In the revised version, the theoretical basis for the classification of musical emotions has been added in section 2.4.1, Lines 215-228, as follows:
“There have been significant attempts at characterizing musical emotions perceived by listeners and describing the affective value and the expressiveness of music (Zentner et al., 2008; Hevner, 1936). Both discrete classification model and continuous dimensional model are commonly used for musical emotion classification (Zentner et al., 2008). In terms of discrete classification, based on the compilation of music-relevant affect terms, adjectives are classified into groups; each group is treated as a whole, representing one type of feeling tone. For instance, Hevner (1936) presented eight synonym clusters to describe emotional perception of music, including happiness, gracefulness, sereneness, dreaminess, sadness, dignity, vigorousness, and excitement. Meanwhile, musical emotions can be described as vectors; typical dimension models include valence-arousal (VA) and pleasant-arousal-dominance (PAD) models. Further, musical emotion characteristics can be classified based on musical dimensions, such as pitch height, loudness, timbre, tempo, and intensity (Wieczorkowska et al., 2006; Trohidis et al., 2011). For instance, referring to the Plutchik 3-D Circumplex Model, a pleasure-arousal (PA) model of musical emotion classification based on register, tempo, and dynamic was presented (Huang et al., 2008).”
The following references were also added:
Hevner K. Experimental studies of the elements of expression in music. American J Psychology, 1936, 48(2):246-268.
Huang Siqi, Zhou Li, Liu Zhentao, Ni Shan, He Jingxian. Empirical Research on a Fuzzy Model of Music Emotion Classification based on Pleasure-Arousal Model. The 37th Chinese Control Conference, pp. 3239-3244, Wuhan, China, July 25-27, 2018.
Trohidis, K., Tsoumakas, G., Kalliris, G., Vlahavas, I., 2011. Multi-label classification of music by emotion. EURASIP Journal on Audio, Speech, and Music Processing 4. https://doi.org/10.1186/1687-4722-2011-426793.
Wieczorkowska, A., Synak, P., Raś., Z.W., 2006. Multi-label classification of emotions in music. In: Kłopotek M.A., Wierzchoń S.T., Trojanowski K. (eds) Intelligent Information Processing and Web Mining. Advances in Soft Computing, vol 35. Springer, Berlin, Heidelberg. https://doi.org/10.1007/3-540-33521-8_30.
Zentner, M, D. Grandjean, and K. R. Scherer. Emotions evoked by the sound of music: characterization, classification, and measurement. Emotion 8.4(2008):494-521.
The selection of experimental materials in the tempo part of the paper is based on the average BPM value of the selected music clips. Does it take into account the tempo fluctuations in the music? For example: the average BPM value of music in the first 15 seconds is 30, the average BPM value of music in the last 15 seconds is 170, and the average BPM value of the entire song is 100.
In the revised version, we added about the tempo fluctuations in music excerpts in section 2.4.2, Line 262-263, as follows:
“Additionally, the melodic structures of the chosen music excerpts are generally regular, with little tempo fluctuations; therefore, the average BPM could be used in tempo analysis.”
The paper only studies the influence of medium tempo (average BPM value = 108.66) and fast (average BPM value = 137) on the emotion of communication, and draws the conclusion that the tempo of music has little effect on emotion. This study lacks the slow tempo. The empirical study of the effects of communication emotions is not comprehensive enough and the conclusions are one-sided. Consider adding experimental music samples for additional research.
In the revised version, the results of the slow tempo have also been added in section 3.3.
Reviewer 4 Report
This manuscript studied the effects of music on communication emotion including the evaluation of the music, d-values of pleasure, arousal and dominance, in terms of sound pressure level, musical emotion and tempo. It has the good value and the conclusion can be used for enhance the communicatin emotion.
1) Line 277, 'Figure 2.ChartoftheExperiment' need to revised;
2) Figures are not clear, need to revise, online version can have the color figures.
Author Response
This manuscript studied the effects of music on communication emotion including the evaluation of the music, d-values of pleasure, arousal and dominance, in terms of sound pressure level, musical emotion and tempo. It has the good value and the conclusion can be used for enhance the communicatin emotion.
- Line 277, 'Figure 2.ChartoftheExperiment' need to revised;
In the revised version, this has been changed to“Figure 2. Chart of the Experiment”.
2) Figures are not clear, need to revise, online version can have the color figures.
All Figures have been revised for clarity.
Reviewer 5 Report
This manuscript investigates the effects of music sounds on emotions in communication. A laboratory experiment was conducted to assess the influences of background music sounds by varying sound pressure levels (SPLs), musical emotions, and tempos. Overall, the topic of this study is interesting and the approach is reasonable. However, there are some unclear points, particularly, on data analysis. Specific comments are asl follows:
- Section Introduction (Line 58): “As for the acoustic environment, studies have mainly focused on negative emotions caused by noise (Park et al., 2018).” The reference (Park et al., 2018) focuses on footstep noises in residential buildings. Thus, I recommend citing other references. For instance, Marquis-Favre, C., Premat, E., & Aubree, D. (2005). Noise and its effects - A Review on Qualitative Aspect of Sound. Part II : Noise and Annoyance. Acta Acoustica United With Acoustica, 91, 626–642.
- Section Introduction (Lines 58-64): “some studies show that air traffic noise seemed to be more annoying than road and rail transport noise (Eggermont, 2014). For example, children’s anxiety was increased by exposure to chronic aircraft noise (Lee et al., 2008). In terms of noise generated by human movement, footstep noise in residential areas can increase annoyance (Park et al., 2018).” The literature reviews are too specific and not directly related to the topics of this manuscript.
- Section Introduction (Lines 64-65): “However, there is a lack of research on positive effects of the acoustic environment on emotion.” There have been many studies on the positive effects of the acoustic environment, particularly, in the field of the soundscape. Thus, this sentence should be revised.
- Section Introduction (Line 105): “Second, how does musical emotion effect communication emotion?” à “Second, how does musical emotion affect communication emotion?”
- Section 2.3 Questionnaire survey (Line 194): “10-point scale (0: no effect/ 1 to 10: positive/ -1 to -10: negative).” Is this 21-point scale?
- Section 3. Results: Table 5 shows the effect size of music properties on emotions. In fact, reporting the effect sizes in the table is not a useful way without explaining statistical information (i.e., statistical methods, degree of freedom, and F/t/p-values). In addition, the values in Table 5 are reported in the text. Thus, it would be better to remove the table in the manuscript to reduce redundancy.
- Section 3.1 Effect of SPL of Music on Communication Emotion: As shown in Figure 3, the percentages of “no effect”, “positive effect” or “negative effect” are calculated. In general, frequency analysis is used for categorical or ordinal scale. Why and how do the authors analyze the percentage of the selection although the numerical rating scale was adopted?
- Statistical report: Throughout the manuscript, the statistical values, namely, statistical methods, degree of freedom, and F/t/p-values should be properly reported.
- There are two discussion sections: Sections 3.4 Discussion and 4. Discussion.
Author Response
This manuscript investigates the effects of music sounds on emotions in communication. A laboratory experiment was conducted to assess the influences of background music sounds by varying sound pressure levels (SPLs), musical emotions, and tempos. Overall, the topic of this study is interesting and the approach is reasonable. However, there are some unclear points, particularly, on data analysis. Specific comments are as follows:
Section Introduction (Line 58): “As for the acoustic environment, studies have mainly focused on negative emotions caused by noise (Park et al., 2018).” The reference (Park et al., 2018) focuses on footstep noises in residential buildings. Thus, I recommend citing other references. For instance, Marquis-Favre, C., Premat, E., & Aubree, D. (2005). Noise and its effects - A Review on Qualitative Aspect of Sound. Part II : Noise and Annoyance. Acta Acoustica United With Acoustica, 91, 626–642.
In the revised version, the reference has been replaced, in Lines 67-70, as follows:
“On the other hand, there are also many studies about negative emotions induced by noise. For instance, some studies suggest that the relationship between annoyance and noise is also related to some psychoacoustic indices, such as loudness, sharpness, roughness, impulsiveness, and so on (Marquis-Favre et al., 2005).”
The following reference is also added:
Marquis-Favre, C., Premat, E., & Aubree, D. (2005). Noise and its effects - A Review on Qualitative Aspect of Sound. Part II : Noise and Annoyance. Acta Acoustica United With Acoustica, 91, 626–642.
Section Introduction (Lines 58-64): “some studies show that air traffic noise seemed to be more annoying than road and rail transport noise (Eggermont, 2014). For example, children’s anxiety was increased by exposure to chronic aircraft noise (Lee et al., 2008). In terms of noise generated by human movement, footstep noise in residential areas can increase annoyance (Park et al., 2018).” The literature reviews are too specific and not directly related to the topics of this manuscript.
In the revised version, these sentences have been deleted.
Section Introduction (Lines 64-65): “However, there is a lack of research on positive effects of the acoustic environment on emotion.” There have been many studies on the positive effects of the acoustic environment, particularly, in the field of the soundscape. Thus, this sentence should be revised.
In the revised version, information about soundscape literature has been added in the INTRODUCTION, Lines 56-67, as follows:
“...On the other hand, a cold or hot environment can elicit negative emotions, which are distracting and reduce performance efficiency (Seppanen and Fisk, 2004). As for the acoustic perception, the term ‘soundscape’ was coined by Schafer, who defined it as a sonic environment, with an emphasis on the way it is perceived and understood by individuals or society (Brown et al., 2011). In 2014, the International Organization of Standardization developed a broader definition of soundscape: acoustic environment as perceived or experienced and/or understood by a person (ISO, 2014). Assessment of soundscape is part of sensory aesthetics research, and the aesthetic response of surroundings is considered to be a mix of high pleasure, excitement, and relaxation (Lang, 1988; Nasar, 1989). In terms of positive effects, a study suggested that nature- and culture-related sounds, such as fountains, birds singing, bells, and music from clock, which induce tranquil and pleasant feelings, are preferred in urban squares, as opposed to artificial sounds (Yang and Kang, 2005).”
The following references have also been added:
Brown, A.L., Kang, J., Gjestland, T., 2011. Towards standardization in soundscape preference assessment. Appl. Acoust. 72 (6), 387–392.
ISO 12913-1, 2014. Acoustics–Soundscape–Part 1: Definition and Conceptual Framework.
Lang, J. (1988) Symbolic aesthetics in architecture: toward a research agenda, in: J. L. Nasar (Ed.)Environmental Aesthetics (Cambridge: Cambridge University Press).
Nasar, J. L. (1989) Perception, cognition, and evaluation of urban places, in: I. Altman & E. H. Zube(Eds) Public Places and Spaces (New York: Plenum Press).
Yang, Wei , and J. Kang . "Soundscape and Sound Preferences in Urban Squares: A Case Study in Sheffield." Journal of Urban Design 10.1(2005):61-80.
Section Introduction (Line 105): “Second, how does musical emotion effect communication emotion?” à “Second, how does musical emotion affect communication emotion?”
In the revised version, “effect” has been changed to “affect”.
Section 2.3 Questionnaire survey (Line 194): “10-point scale (0: no effect/ 1 to 10: positive/ -1 to -10: negative).” Is this 21-point scale?
In the revised version, “10-point scale” has been changed to “21-point-scale”
Section 3. Results: Table 5 shows the effect size of music properties on emotions. In fact, reporting the effect sizes in the table is not a useful way without explaining statistical information (i.e., statistical methods, degree of freedom, and F/t/p-values). In addition, the values in Table 5 are reported in the text. Thus, it would be better to remove the table in the manuscript to reduce redundancy.
In the revised version, this Table has been deleted.
Section 3.1 Effect of SPL of Music on Communication Emotion: As shown in Figure 3, the percentages of “no effect”, “positive effect” or “negative effect” are calculated. In general, frequency analysis is used for categorical or ordinal scale. Why and how do the authors analyze the percentage of the selection although the numerical rating scale was adopted?
In the revised version, it has been added in section 3.1, Line 340-341, as follows:
“To determine the distribution of participants in terms of the three levels of musical evaluation—positive, negative, and no effect— percentages were also analysed..”
Statistical report: Throughout the manuscript, the statistical values, namely, statistical methods, degree of freedom, and F/t/p-values should be properly reported.
In the revised version, F/ t/ p values have been added in the RESULTS.
There are two discussion sections: Sections 3.4 Discussion and 4. Discussion.
In the revised version, the subheadings have been revised to 4. Discussion and 5. Conclusions.
Round 2
Reviewer 3 Report
All the issues I gave have been well addressed.
Author Response
N/A
Reviewer 5 Report
The authors successfully have revised the manuscript according to the reviewer's comments. I believe that the present manuscript is qualified to be published in IJERPH.
Author Response
N/A